# Regulatory Role of Trehalose Metabolism in Cold Stress of *Harmonia axyridis* Laboratory and Overwinter Populations

**Sijing Wan [1,2], Jianyun He [1,2], Lei Chao [1], Zuokun Shi [1], Shasha Wang [1], Weidong Yu [3], Zhen Huang [3], Su Wang [4], Shigui Wang [1,*] and Zhijun Zhang [2,*]**

1   Hangzhou Key Laboratory of Animal Adaptation and Evolution, College of Life and Environmental Sciences, Hangzhou Normal University, Hangzhou 310036, China
2   Institute of Plant Protection and Microbiology, Zhejiang Academy of Agricultural Sciences, Hangzhou 310021, China
3   Zhejiang Dingyi Biotechnology Corporation, Quzhou 324100, China
4   Institute of Plant and Environment Protection, Beijing Academy of Agriculture and Forestry Sciences, Beijing 100089, China
*   Correspondence: sgwang@hznu.edu.cn (S.W.); zhijunzhanglw@hotmail.com (Z.Z.)

**Abstract:** Trehalose is a non-reducing disaccharide that plays a key role in the response to cold and other environmental stressors in insects. *Harmonia axyridis* (Pallas) is an important natural predator of insect pests and has become a cosmopolitan invasive species, causing negative ecological impacts worldwide. In this study, the relative survival ability, trehalose and glycogen contents, trehalose activity and trehalose metabolism-related gene-expression profiles over a cold storage period were compared in a natural overwintering population and an indoor laboratory (experimental) population. Yellow adults were dominant in the overwintering population. The survival rate of the overwintering population during the cold storage period was higher than that of the experimental population after the fifth day. The contents of trehalose and glycogen in the overwinter population were higher than those of the experimental population during cold storage. Trehalose and glycogen contents initially increased and then decreased in the overwinter population, but decreased consistently over the cold storage period in the experimental population. Nevertheless, trehalose levels were relatively higher during the cold storage period in the overwinter population, with higher expression of *TPS* and *TRE* and trehalase activity. More importantly, the experiment showed that yellow adults have a better ability to regulate trehalose metabolism under cold storage compared to black adults. A strong resilience against cold stress and long-term cold storage ability could contribute to a better understanding of the invasiveness of *H. axyridis*.

**Keywords:** *Harmonia axyridis* (Pallas); cold hardiness; trehalose; trehalase; RT-qPCR

## 1. Introduction

The Asian lady beetle (or ladybird) *Harmonia axyridis* (Pallas) (order Coleoptera, family Coccinellidae) has a strong ability for predation on multiple important crop pests, thus, it represents an important natural insect predator, notably for biological control [1–4]. Although *H. axyridis* has been widely used in agricultural production worldwide, it has now spread widely in forest, farmland, grassland and other ecosystems, becoming a cosmopolitan invasive species with negative ecological impacts [5–7]. For example, some surveys showed that *H. axyridis* became a dominant coccinellid in some habitats, leading to a sharp decline in native species such as *Adalia bipunctata*. In addition, it can directly and indirectly damage some fruits such as raspberries and grapes [6]. Therefore, this insect has attracted increasing research attention in recent years, e.g., focusing on its viability, invasive ability, adaptation ability, and influence on other insects with the same ecological niche [2,7–10]. Adult *H. axyridis* are well known for their elytra of rich colors with a black (melanic) or yellow (non-melanic) background and inlaid black or red dots [11,12], regulated by the expression of a series of alleles [13]. The four major color-regulating alleles

are known as conspicua, spectabilis, axyridis, and succinea [13]. Environmental factors may lead to ladybird elytral diversity [14,15]. Color ratios vary in different areas and in different seasons in the same area [16,17]. Some studies have suggested that this seasonal phenotypic plasticity may be related to cold hardiness and overwintering strategies [18].

Trehalose is a non-reducing disaccharide, which is considered a "blood sugar" because of its important functions in insect growth and development [19]. Trehalose can accumulate in response to environmental stress, including desiccation, cold, oxidation and anoxia [19–22], and also plays a key role in the insect anti-stress response as an intermediate product [23]. Trehalose can be hydrolyzed into two glucose molecules by trehalase (TRE) or via inter-conversion with another sugar under cold, starvation and other stress conditions. For example, glycogen can transform into trehalose through the trehalose and glycogen metabolism pathway [24]. Two forms of trehalase, soluble trehalase (TRE1) and membrane-bound trehalase (TRE2), have been identified to date and cloned in many insect species [23,25,26]. Insect trehalose can now be synthesized by a variety of organisms and its clear roles in energy metabolism have been defined [27]. In insects, trehalose is mainly synthesized by trehalose-6-phosphate synthase (TPS) and trehalose-6-phosphate phosphatase (TPP) [27]. However, both trehalose and its biosynthesis genes can only be cloned in a few insect species [28]. TPS catalyzes the transfer of glucose from uridine diphosphate (UDP)-glucose to glucose 6-phosphate to generate trehalose 6-phosphate (T6P), and then TPP catalyzes the dephosphorylation of T6P to form trehalose [29]. UDP-glucose is also the substrate for glycogen synthesis by glycogen synthase (GS); glycogen can be hydrolyzed to glucose-1-phosphate by glycogen phosphorylase (GP) and glucose-1-phosphate can be inter-converted to glucose-6-phosphate by phosphoglucomutase (PGM).

The winter temperature is a major determinant of the distribution of insect species. The cold tolerance strategies of insects are generally divided into two major categories: freeze tolerance and freeze intolerance [30]. Insects that exhibit cold acclimation before winter in nature acquire increased cold tolerance to survive in the low-temperature environment of winter [31]. Studies have indicated that even moderate cold acclimation before low-temperature stress can increase the survival rate, reduce the lethal temperature, prolong the half lethal time and decrease the super-cooling point of insects [32]. As the insect "blood sugar", trehalose is suggested to play a key role in adapting to cold stress and starvation in *H. axyridis* [21]. In fact, total sugars were found to decrease significantly over time when overwintering adults were stored in a cold condition [33]. Although *H. axyridis* exhibits stronger cold tolerance in winter, the precise physiological and metabolic changes contributing to this cold hardiness are unclear, including the involvement of trehalose or glycogen. Therefore, in this study, we evaluated the trehalose and glycogen contents in the yellow and black forms of *H. axyridis* adults in the wild, and explored the changes in the expression of genes in the trehalose and glycogen pathways over time in an overwintering population and an experimental population during cold storage in the laboratory. The experimental population was reared and maintained indoors under laboratory conditions, and the overwintering population was collected from the wild just prior to the winter season (end of September/early October). We compared the changes in the ratio of yellow adults and the survival rate over the cold storage time. These findings can provide new insight into the potential molecular and physiological mechanism of cold resistance in *H. axyridis*, which can in turn offer a strategy for its more efficient use as a natural predator while preventing its invasive spread.

## 2. Materials and Methods

### 2.1. Establishment of Experimental Insects

Adult *H. axyridis* were collected in a wheat field on the campus of Beijing Academy of Agriculture and Forestry Sciences in Beijing, China, in April 2021. The healthy beetles were held in aluminum frame cages (40.0 × 40.0 × 45.0 cm) covered with nylon mesh, with 30 pairs per cage, and fed *Megoura crassicauda* in a climate-controlled growth chamber (Sanyo, MH351, Osaka, Japan). To establish the experimental colony, the

*H. axyridis* in the climate chamber were able to mate freely. The *H. axyridis* were provisioned daily with fresh *M. crassicauda* on bean shoots. Eggs were collected every 12 h and placed in plastic hatching containers. Hatched larvae were transferred to large aluminum frame cages (65.0 × 48.0 × 48.0 cm) with 150 individuals per cage and fed until pupation. Emerged adults were divided into two groups according to the elytra color: black or yellow. Adults of each group were maintained in plastic boxes (as described above), with 30 beetles per box, and fed fresh *M. crassicauda* daily for at least 10 days prior to use in the low-temperature storage experiments. The chambers were set to a constant temperature of 25.0 ± 1.0 °C, with a relative humidity of 60–70% and a 16:8 h light:dark photoperiod under a light intensity of 600 lux. All experimental insects were maintained under the same conditions as the stock colony.

### 2.2. Distribution of Elytra Color of H. axyridis in a Natural Overwintering Population

*H. axyridis* were mainly collected in Laoyeling at the experimental forest farm of Northeast Forestry University from Mao'er Mountain in Heilongjiang Province (N45°20′–45°25′; E127°30′–127°34′) in late September or early October from 2013 to 2021. The overwintering ladybirds often gather in the crevices of rocks and the roots of trees. Combined with Mao'er Mountain topography, near the ecological positioning station, nine typical sunny sampling points with an area of about 25 m × 25 m were randomly selected for this experiment, and the tree species in the sample site were typical of broad-leaved forests in this area. The adults were transported to the laboratory and killed by being placed in a dryer. The insects were then divided randomly into three replicate groups with at least 2000 individuals per group, and the proportion of melanic (black) or succinic (yellow) individuals was calculated according to the background coloration. The larvae were sealed with gauze and stored outdoors.

### 2.3. Low-Temperature Storage Treatment and Survival Rate Analysis

A total of 200 black and yellow adults from an overwintering colony of *H. axyridis* collected from Mao'er Mountain in late September and 200 black and yellow adults (ten days of eclosion) from the experimental population were subjected to low-temperature storage at 5 °C in a refrigerator. At 0, 5, 10, 15, 20, 40 and 60 days, 3–5 live adults (♀:♂ = 1:1) were removed and stored at −80 °C until analysis for trehalose/glycogen content, enzyme activity and mRNA quantification. The experiment was repeated three times.

Simultaneously, a separate group of 180 yellow or black adults of the overwintering and experimental populations was selected for storage at 5 °C, and survival rates were calculated at 0, 5, 10, 15, 20, 40 and 60 days. The experiment was repeated three times.

### 2.4. Measurement of Trehalose and Glycogen Contents under Cold Stress

Determination of sugar content was performed using the anthrone method. Trehalose content was measured as described previously [21]. Three adults were placed in a 5 mL Eppendorf tube. After adding 500 μL 20 mM phosphate buffered saline (PBS, pH 6.0), tissues were homogenized at 0 °C (TGrinder OSE-Y20 homogenizer, TIANGEN BIOTECH CO., LTD., Beijing, China), and this was followed by sonication for 30 min (VCX 130PB, Sonics, Liège, Belgium). Homogenates were centrifuged at 12,000× $g$ at 4 °C for 10 min after adding 2.5 mL PBS. Precipitates were removed and aliquots of supernatant were assayed to determine the protein content using a protein dye-binding method (Beyotime, Shanghai, China)) with bovine serum albumin as the standard. Then, 1 mL supernatant was added to a 1.5 mL tube and boiled, after which the solution was centrifuged at 12,000× $g$ for 10 min to remove any residual protein. The supernatant was divided into two tubes: one was directly subjected to a glycogen content assay, while the other was processed for measurement of trehalose. For trehalose, firstly, we hydrolyzed glycogen into glucose in sulfuric acid ($H_2SO_4$) heating conditions and then decomposed the total glucose in solutions under alkaline conditions. Thus, the supernatant contained trehalose without other carbohydrates or proteins. The details of the trehalose test are as follows.

To test trehalose content, aliquots of supernatant (100 μL) were put into a 1.5 mL tube, 100 μL 1% $H_2SO_4$ was added, and the tube was incubated in 90 °C water for 10 min to hydrolyze glycogen, after which it was cooled on ice for 3 min. Then, the supernatant was again incubated in 90 °C water for 10 min after adding 100 μL of 30% potassium hydroxide solution to decomposed glucose. Then, 4 volumes of 0.2% (M/V) anthrone (Sigma, St. Louis, MO, USA) in 80% $H_2SO_4$ solution were added after it was cooled on ice for 3 min, and the supernatant was boiled for 10 min. After cooling, 200 μL reaction solution was placed into a 96-well plate and the absorbance at 620 nm was determined using a SpectraMax M5 (Molecular Device, Sunnyvale, USA). Trehalose content was calculated based on a standard curve and was compared with the total protein content. Finally, the result was expressed as mg trehalose per g total protein.

Glycogen content was measured as described previously [34]. A total of 200 μL of supernatant (described above) was incubated for 4 h at 37 °C in the presence of 40 μL (1 U) amyloglucosidase (EC 3.2.1.3, Sigma, St. Louis, MO, USA) diluted in 100 mM sodium acetate (pH 5.5) to hydrolyze glycogen. The amount of glucose generated from glycogen was determined using a Glucose Assay Kit (GAGO20-1KT, Sigma, St. Louis, MO, USA) following the manufacturer's instructions. Controls were prepared in the absence of the enzyme, and the amount of glycogen was calculated as follows: total glucose minus endogenous glucose, then divided by the total protein. Finally, the result was expressed as mg glucose per g total protein.

### 2.5. Soluble Trehalase and Membrane Bound Trehalase Activity Assay

Trehalase activity was determined according to a previously described method [35]. The amount of glucose generated from glycogen was determined using a Glucose Assay Kit (GAGO20-1KT, Sigma, St. Louis, MO, USA) following the manufacturer's instructions.

### 2.6. RT-qPCR Analysis

Total RNA was isolated and 1 μg total RNA was used for the synthesis of first strand complementary DNA (cDNA) using a PrimeScript RT® with gDNA Eraser kit (TaKaRa, Dalian, China). The mRNA expression level of seven trehalase genes, *TPS*, *GS* and *GP* was estimated with qPCR using a Bio-Rad CFX96™ system (Bio-Rad, Hercules, CA, USA). Specifically, the 20 μL total reaction volume contained 1 μL cDNA sample, 1 μL (10 μmol/μL) of each primer, 7 μL RNase-free and DNase-free water and 10 μL SsoFastTM EvaGreen® Supermix. The template was replaced with $H_2O$ as a negative control and the "house-keeping" gene Harp49 (*H. axyridis* ribosomal protein 49 gene) was used as the internal reference [36,37]. Primers are listed in Table 1. The cycling parameters were 94 °C for 5 min for initial denaturation, followed by 40 cycles at 94 °C for 15 s and 59 °C for 30 s; melting curve analysis at 65 °C to 95 °C was performed. Data were analyzed using the ΔΔCt relative quantitative method [38].

### 2.7. Statistical Analysis

Insects were randomly allocated into different groups with three replicates for each treatment. Results were expressed as the mean ± standard deviation (SD) or the mean ± standard error (SE) of independent replicates (n ≥ 3). The data were analyzed using IBM SPSS statistics v20 software. Statistical significance was defined as $p < 0.05$. Tukey's test of one-way ANOVA was performed to test the significance of differences among treatments. All figures and tables were produced using Microsoft Office 2013 and SigmaPlot 10.0 software.

**Table 1.** Primers used in real-time polymerase chain reaction.

| Gene Name | Primer Name | Nucleotide Sequences (5′-3′) | GenBank Number |
|---|---|---|---|
| *Treh1-1* | HaTreh1-1QF | CTTCGCCAGTCAAATCGTCA | HM056038 |
| | HaTreh1-1QR | CCGTTTGGGACATTCCAGATA | |
| *Treh1-2* | HaTreh1-2QF | TGACAACTTCCAACCTGGTAATG | FJ501961 |
| | HaTreh1-2QR | TTCCTTCGAGACATCTGGCTTA | |
| *Treh1-3* | HaTreh1-3QF | ACAGTCCCTCAGAATCTATCGTCA | JX514372 |
| | HaTreh1-3QR. | GGAGCCAAGTCTCAAGCTCATC | |
| *Treh1-4* | HaTreh1-4QF | TTACTGCCAGTTTGATGACCATT | KP318742 |
| | HaTreh1-4QR | CATTTCGCTAATCAGAAGACCCT | |
| *Treh1-5* | HaTreh1-5QF | TGATGATGAGGTACGACGAGAA | KX349223 |
| | HaTreh1-5QR | GTAGCAAGGACCTAACAAACTGC | |
| *Treh2-1* | HaTreh2-1QF | TTCCAGGTGGGAGATTCAGG | KX349224 |
| | HaTreh2-1QR | GGGATCAATGTAGGAGGCTGTG | |
| *Treh2-2* | HaTreh2-2QF | CAATCAGGGTGCTGTAATGTCG | KX349225 |
| | HaTreh2-2QR | CGTAGTTGGCTCATTCGTTTCC | |
| *TPS* | HaTPS-QF | GACCCTGACGAAGCCATACC | FJ501960 |
| | HaTPS-QR | AAAGTTCCATTACACGCACCA | |
| *GP* | HaGP-QF | GCTGAAGCCCTCTACCAACT | NC059507 |
| | HaGP-QR | CGCCGTACTCGTATCTTATGC | |
| *GS* | HaGS-QF | CCCTTAGGATCGGATGTTCTC | NC059503 |
| | HaGS-QR | CACCAGCCATCTCCCAGTT | |
| *Harp49* | Harp49-qF | GCG ATC GCTATGGAAAAC TC | AB552923 |
| | Harp49-qR | TACGATTTTGCATCAACAGT | |

## 3. Results

### 3.1. Distribution of Yellow and Black H. axyridis in the Overwinter Population

The numbers of yellow and black *H. axyridis* among a natural overwinter population collected in late September from Mao'er Mountain (Heilongjiang, China) were collected from 2013 to 2021. As shown in Figure 1, yellow adults were dominant in all three batches, with a proportion of 88.68%, 88.75%, and 88.85%.

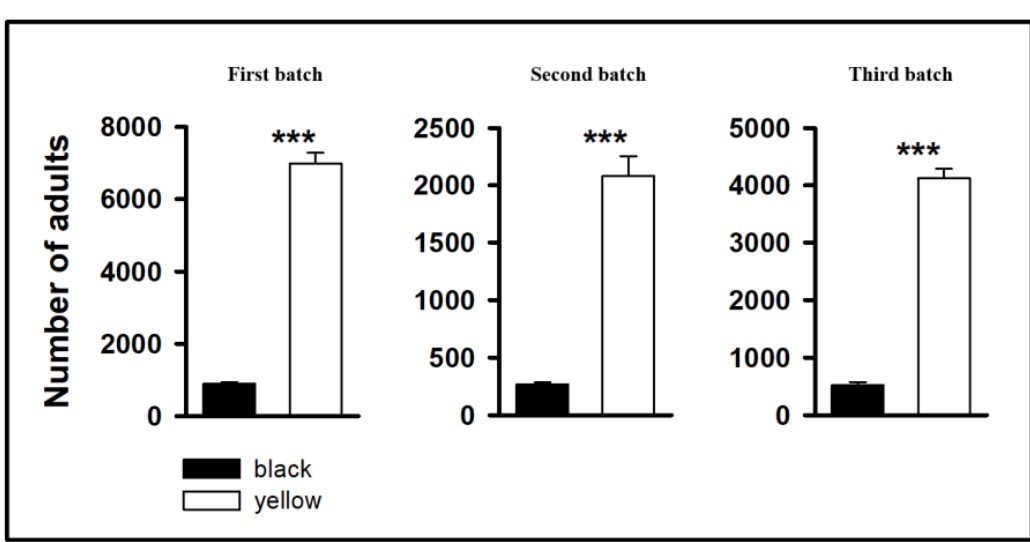

**Figure 1.** Numbers of yellow and black *H. axyridis* adults observed in nature in Heilongjiang province, China. Each bar depicts the mean (+SD) of different sampling years before the winter (September/October) in each of these years. Asterisks indicate significant differences between absolute numbers in each year (*t*-test, *** $p \leq 0.001$).

### 3.2. Changes in Trehalose and Glycogen Contents under Cold Stress

In the overwinter population, the change trends of trehalose and glycogen levels during cold storage (5 °C) were similar between the yellow and black *H. axyridis* adults.

The content of trehalose decreased significantly at 5 days, followed by a slight increase at 10 days and a subsequent significant decrease from 10 to 60 days in the overwinter black population; thus, the trehalose content was highest at day 0 and lowest at day 60 of cold storage (Figure 2(Ai)). The content of trehalose in the yellow population did not significantly change during the first 15 days of cold-stress storage and then decreased significantly, with no further changes from day 20 to day 60; the highest level was at day 0 and the lowest level was at day 40. The content of trehalose in yellow adults was higher than that of black adults at 5 days and 15 days, whereas this trend shifted at 20 and 40 days for the overwinter population (Figure 2(Ai)). The content of glycogen showed an increasing trend from 0 to 20 days, with a relatively lower level at 40 and 60 days in the black population; the highest level was detected at 10 days and the lowest level was detected at 60 days (Figure 2(Bi)). The content of glycogen in the yellow population did not significantly change during the first 15 days of cold-stress storage, although it increased from 5 to 15 days and then decreased significantly from 20 to 40 days; the lowest level was detected at day 60 and the highest level was detected at day 10 (Figure 2(Bi)).

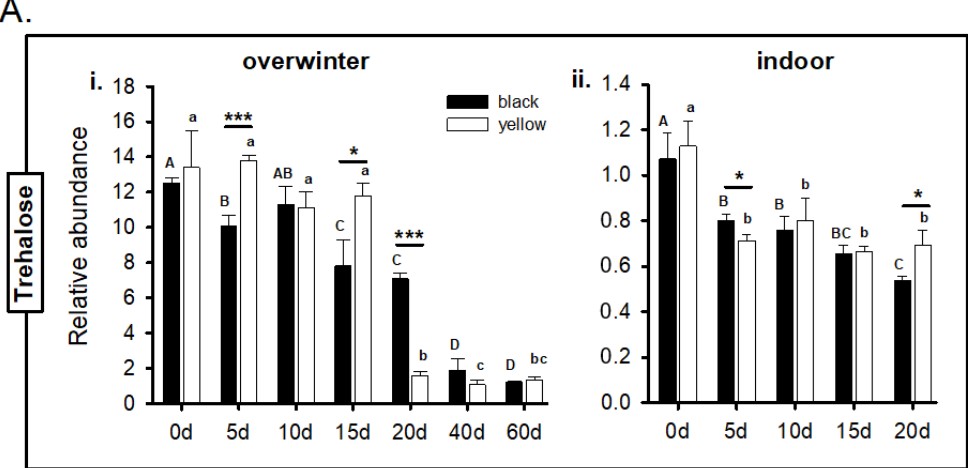

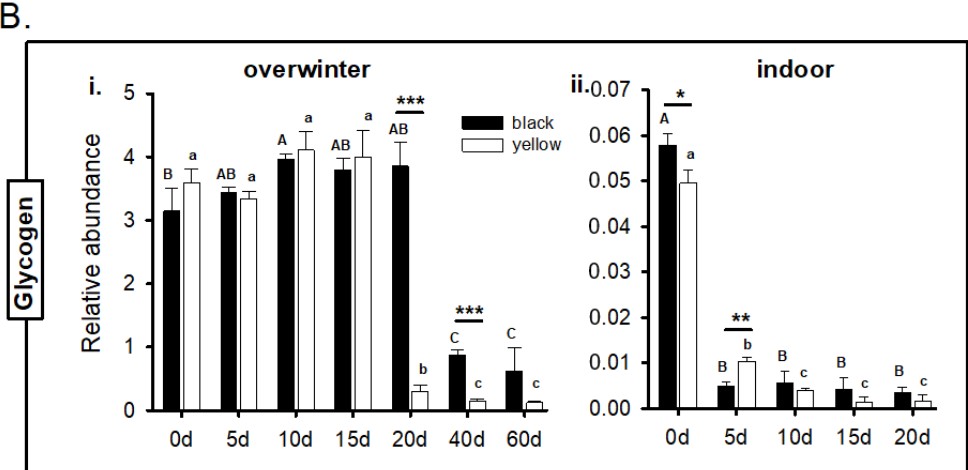

**Figure 2.** Changes in the contents of (**A**) trehalose and (**B**) glycogen in yellow (non-melanic) and black (melanic) *H. axyridis* adults of overwinter (i) and experimental (indoor) (ii) populations during cold storage (0–60 days and 0–20 days, respectively). Bars represent the mean (+SD) of three replicate experiments. The content of trehalose or glycogen at 0 d cold storage was used as the control group. Bars with different letters indicate significant differences (one-way ANOVA, $p \leq 0.05$). Asterisks indicate significant differences between yellow and black adults in the same storage period (*t*-test, *** $p \leq 0.001$; ** $p \leq 0.01$; * $p \leq 0.05$).

In the experimental (indoor laboratory) population, the contents of trehalose and glycogen showed a decreasing trend over time under cold stress. The content of trehalose in black adults decreased significantly from 0 to 20 days (Figure 2(Aii)). The trehalose content of yellow adults was higher at 0 days than during storage, but did not decrease significantly over time; the lowest level was detected at day 15 of cold storage (Figure 2(Aii)). The content of glycogen rapidly decreased significantly for both yellow and black adults at 5 days of cold storage (Figure 2(Bii)); the highest glycogen content for both black and yellow adults was detected at 0 days. There was no significant change in the glycogen content of black adults in the experimental population from 5 to 20 days of storage, whereas there was a decrease in glycogen for the yellow adults during this period (Figure 2(Bii)). The glycogen content of black adults at 20 and 40 days was significantly ($p < 0.001$) higher than that of yellow adults in the overwinter population before cold storage treatment, whereas the opposite trend was found at day 5 of storage in the experimental population of *H. axyridis* adults (Figure 2B).

*3.3. Changes in Trehalase Activity during Cold-Stress Storage*

The utilization of trehalose during cold storage was assessed by measuring the activity of soluble trehalase (TRE1) and membrane-bound trehalase (TRE2) enzymes. In the overwinter population, the TRE1 activity of black adults decreased at 5 days and then increased to reach its highest level at 15 days, which was then maintained up to the end of the 60-day storage period (Figure 3A). By contrast, the TRE1 activity of the yellow adults increased from day 5 to day 20 (the highest level), followed by a significant decrease to the lowest level at day 60 (Figure 3A). Thus, the TRE1 activity of yellow adults at days 5, 10, and 20 of cold storage was higher than that of black adults, whereas the opposite pattern was observed at days 15 and 60 for the overwinter *H. axyridis* population (Figure 3A). Similarly, for the experimental population, the TRE1 activity of the black adults was higher than that of the yellow adults at days 10 and 15 of cold storage (Figure 3B). Compared to the TRE1 activity, the activity of TRE2 showed a slight increasing trend.

In the experimental population, the activity of both trehalases showed a decreasing trend as storage progressed. The TRE1 activity at day 0 decreased significantly to reach its lowest level at 20 days in black adults and 15 days in yellow adults (Figure 3B). Similarly, the highest TRE2 activity was observed at day 0 and decreased to its lowest level at day 20 for both black and yellow adults (Figure 3D). Although the TRE2 activity of black adults was higher than that of yellow adults at 5 and 15 days in the overwinter population (Figure 3C), the TRE2 activity of black adults was lower than that of yellow adults from 10 to 20 days in the experimental population (Figure 3D).

*3.4. Expression of TRE and TPS Genes in the Overwinter Population during Cold Storage*

The mRNA levels of seven *TRE* genes and one *TPS* gene were detected using reverse transcription quantitative polymerase chain reaction (RT-qPCR). The expression levels of the eight genes differed between the black and yellow adults of the overwinter population (Figure 4). The mRNA levels of *TRE1-1*, *TRE1-5* and *TRE2-1* increased during cold storage, with *TRE1-1* and *TRE1-5* reflecting TRE1 activity. The mRNA level of *TRE1-1* in black adults was higher than that of yellow adults at days 10 and 40, whereas the opposite pattern was detected at days 15 and 20. In addition, the expression level of *TRE1-5* in yellow adults was higher than that of black adults at days 5 and 40, with an opposite pattern detected at day 20. The expression level of *TRE2-1* in yellow adults was higher than that of black adults at days 5 and 20, whereas the *TRE2-1* level of black adults was higher than that of yellow adults at days 40 and 60. The expression levels of *TRE1-2* and *TRE1-3* showed a decreasing trend in both black and yellow adults; however, the expression level of *TRE1-2* of yellow adults was lower at 15 days, with the opposite patterns found at days 40 and 60. The expression level of *TRE1-3* in black adults was higher than that of yellow adults at day 10. The expression level of *TRE1-4* was higher in the yellow adults than in the black adults from day 5 to 40. The expression level of *TPS* decreased significantly at 15 days in

yellow adults and 20 days in black adults. Moreover, the *TPS* level was higher in black adults than in yellow adults at days 5 and 20, whereas the opposite pattern was observed at days 15 and 40 of cold storage (Figure 4).

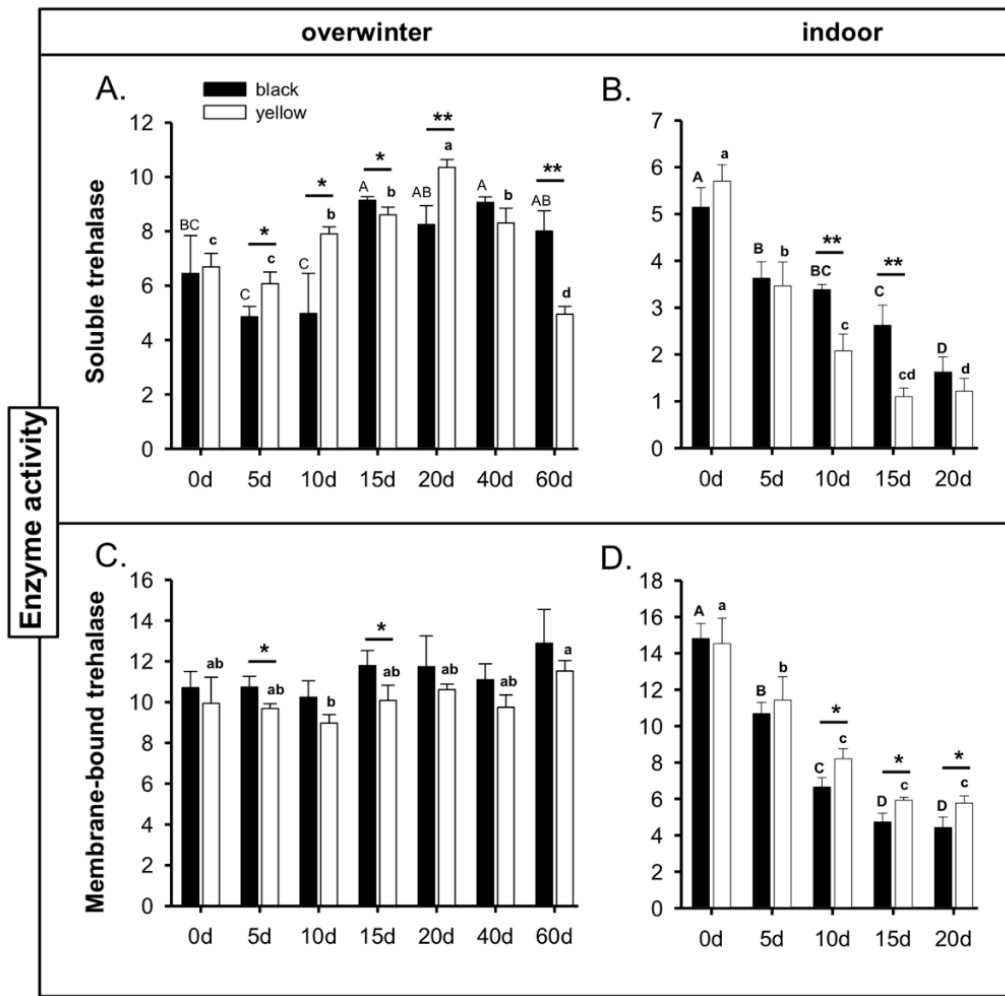

**Figure 3.** Changes in two trehalase enzymes in *H. axyridis* yellow (non-melanic) and black (melanic) adults in the overwinter and experimental (indoor) populations under different cold storage periods (0–60 days and 0–20 days, respectively). (**A**,**B**) Changes of TRE1 during cold storage in overwinter and experimental (indoor) *H. axyridis* populations. (**C**,**D**) Changes of TRE2 during cold storage in overwinter and experimental (indoor) *H. axyridis* populations. Bars represent the mean (+SD) of three replicate experiments. The trehalase activity at 0 d cold storage was used as the control group. Bars with different letters indicate significant differences (one-way ANOVA, $p \leq 0.05$). Asterisks indicate significant differences between yellow and black adults in the same storage period (*t*-test, ** $p \leq 0.01$; * $p \leq 0.05$).

### 3.5. Expression of TRE and TPS Genes in the Experimental Population during Cold Storage

The changes in expression levels of the *TRE* and *TPS* genes showed different trends in the experimental population compared with those detected in the overwinter population. Despite the decrease in trehalase activity from day 0 to 20 of cold storage, the mRNA expression levels of *TRE1-3*, *TRE1-4*, *TRE1-5*, *TRE2-1* and *TRE2-2* increased during the storage period, whereas the levels of *TRE1-1* and *TRE1-2* decreased over time compared with those recorded on day 0 (Figure 5). Specifically, the *TRE1-1* expression level initially decreased significantly and then increased, and the expression levels of yellow adults was higher than that of black adults on days 5, 10 and 20, but not on day 15. The mRNA level of *TRE1-2* was lower on day 20 of storage compared with that recorded on day 0. The *TRE1-3*,

*TRE1-4*, *TRE1-5* and *TRE2-1* expression levels increased significantly from day 0 to 15 and then decreased significantly on day 20. The expression level of *TRE2-2* initially increased at day 5 and then decreased at day 15 of storage (Figure 5). The *TRE1-4* ex-pression level of yellow adults was higher than that of black adults on days 15 and 20, and the same pattern was found for *TRE2-2* at day 20; however, the expression level of *TRE2-1* was significantly higher in black adults than that in yellow adults on day 20. The expression level of *TPS* decreased from day 5 to 10 and was then maintained at a relatively similar level until the end of the storage period, with no differences between yellow and black adults (Figure 5).

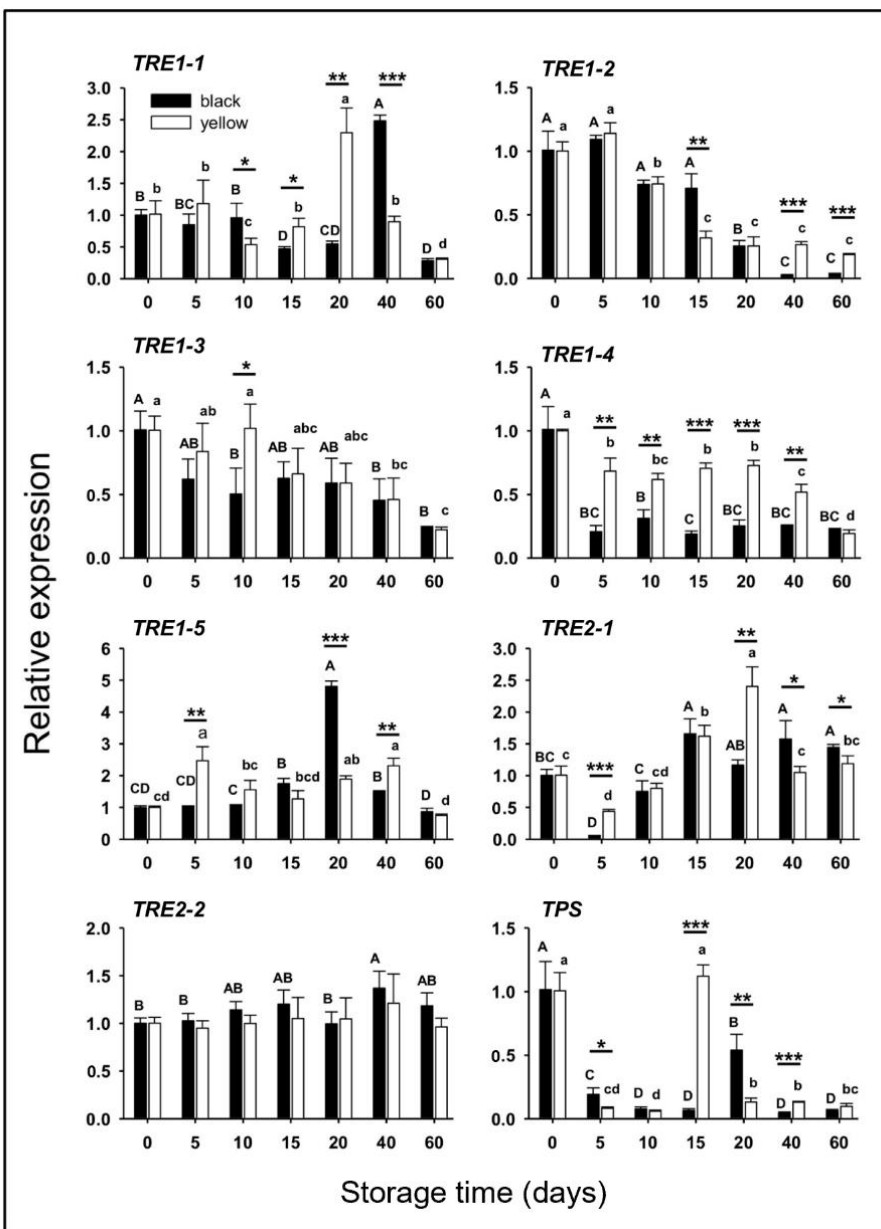

**Figure 4.** Changes in mRNA expression levels of seven trehalase genes and one *TPS* gene in *H. axyridis* yellow (non-melanic) and black (melanic) adults of the overwinter population during cold storage (0–60 days). The relative expression levels of the target genes were calculated using the Harp49 (*H. axyridis* ribosomal protein 49 gene) expression level with RT-qPCR. Bars represent the mean (+SD) of three replicates. The expression level of each gene at 0 d of cold storage was used as the control group. Bars with different letters indicate significant differences (one-way ANOVA, $p \leq 0.05$). Asterisks indicate significant differences between yellow and black adults in the same storage period (*t*-test, *** $p \leq 0.001$; ** $p \leq 0.01$; * $p \leq 0.05$).

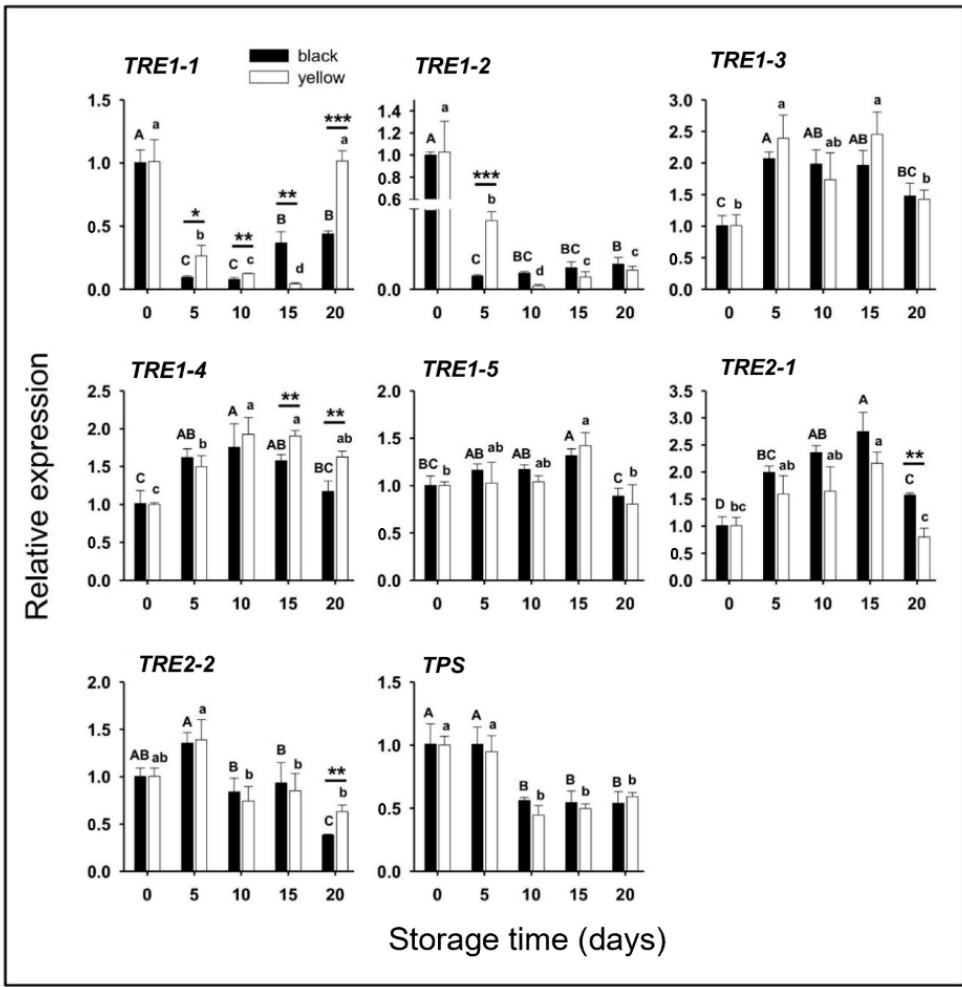

**Figure 5.** Changes in the mRNA expression levels of seven trehalase genes and one *TPS* gene in *H. axyridis* yellow (non-melanic) and black (melanic) adults in the experimental (indoor) population under cold storage (0–20 days). The relative levels of the target genes were calculated according to the Harp49 (*H. axyridis* ribosomal protein 49 gene) expression level with RT-qPCR. Bars represent the mean (+SD) of three replicates. The gene expression level at 0 d of cold storage was used as the control group. Bars with different letters indicate significant differences (one-way ANOVA, $p \leq 0.05$). Asterisks indicate significant differences between yellow and black adults in the same storage period (*t*-test, *** $p \leq 0.001$; ** $p \leq 0.01$; * $p \leq 0.05$).

*3.6. GP and GS Gene Expression during Cold Storage*

The expression levels of the *GP* and *GS* genes, which are related to glycogen synthesis, were also measured during cold storage in both the experimental and overwinter populations. In the overwinter population, the *GP* expression level increased significantly at 20 or 40 days in both yellow and black adults, followed by a significant decrease (Figure 6A). By contrast, in the experimental population, the *GP* expression level increased significantly at day 5, followed by a decrease (Figure 6A). The *GP* expression level in yellow adults was higher than that of black adults at 20 days, with the opposite pattern observed at 40 days in the overwinter population; however, the *GP* expression level was higher in black adults than in yellow adults at day 5 in the experimental population. The expression level of *GS* showed a decreasing trend for both populations, with higher expression levels for yellow adults at days 10, 15 and 60 in the overwinter population (Figure 6B) and at day 10 in the experimental population (Figure 6B).

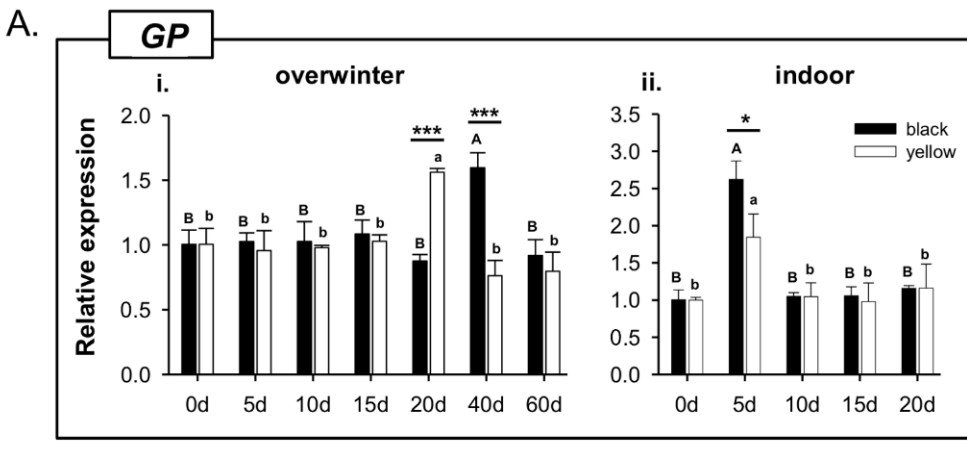

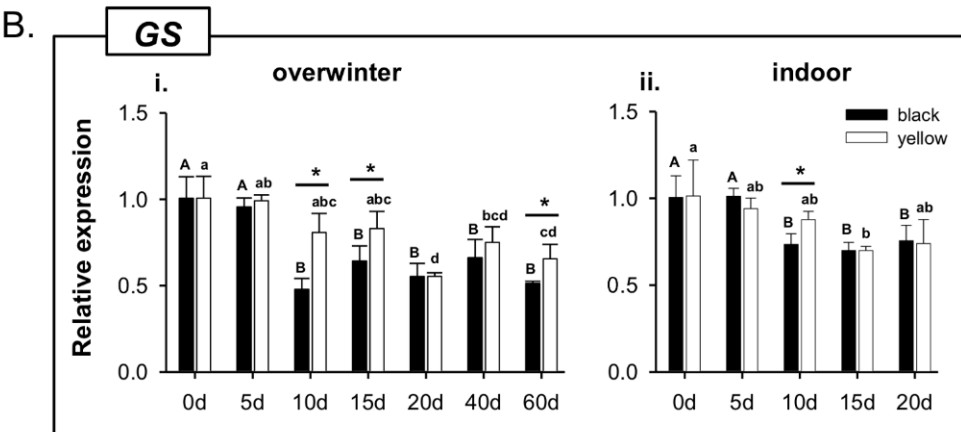

**Figure 6.** Changes in mRNA expression levels of *GP* (**A**) and *GS* (**B**) genes in *H. axyridis* yellow (non-melanic) and black (melanic) adults of the overwinter and experimental (indoor labora-tory) populations under different cold storage periods (0–60 days and 0–20 days, respectively). The relative levels of the target genes were calculated based on the expression level of Harp49 (*H. axyridis* ribosomal protein 49 gene) with RT-qPCR. Bars represent the mean (+SD) of three replicates. The expression level of each gene at 0 d of cold storage was used as the control group. Bars with different letters indicate significant differences (one-way ANOVA, $p < 0.05$). Asterisks indicate significant differences between yellow and black adults in the same storage period (*t*-test, *** $p \leq 0.001$; * $p \leq 0.05$).

### 3.7. Survival Rates of H. axyridis during Cold Storage

The survival rate of the overwinter population was higher than that of the experimental population from days 5 to 40 of cold storage (Figure 7). The adult survival rate of the overwinter population was approximately 65%, whereas that of *H. axyridis* adults stored at 5 °C for 20 days was only approximately 14%. The overall survival rate for the overwinter population was 20%, whereas all of the experimental insects died by the end of the 60-day cold-storage period. These results confirmed that the overwinter population had greater cold resistance than the experimental population.

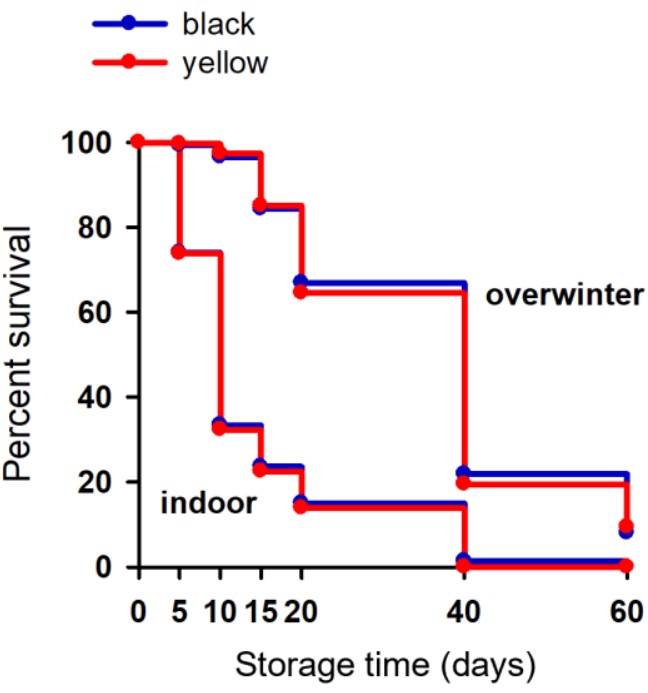

**Figure 7.** Representative survival curves of overwinter and experimental (indoor) black and yellow adults after low-temperature storage for 60 days; n = 180–235 adult beetles. Data are representative of three independent experiments.

## 4. Discussion

Populations of the ladybird *H. axyridis* include adults of both melanic and non-melanic forms [39]. We found that the ratio of non-melanic *H. axyridis* adults in a natural population in Heilongjiang province of northeast China was nearly 90% (Figure 1), demonstrating that yellow is the dominant elytra color for the *H. axyridis* adult population in the winter of this region, which is in line with previous reports [33,40]. In addition, those studies found that some external environmental factors may lead to the diversity found in ladybird coleoptera. The frequency of color spots in ladybird beetles can vary with the seasons [41]. The proportion of color spots of *H. axyridis* also varies at different times. The number of light-colored *H. axyridis* increased in spring and early summer. After summer, the number of black-bottomed *H. axyridis* began to increase, and the number of light-colored *H. axyridis* increased in autumn [33]. Subsequent studies found that the seasonal variation of this color pattern was related to non-random mating selection. In summer, females tended to mate with black-bottomed males, while in autumn, yellow-bottomed males were more popular with females [41]. This seasonal variation may indicate that yellow is a protective color during winter and is related to the temperature change [6,42,43]. Gengzhengcheng and Tan Jiazhen [44] believed that this variation might be related to protective colors. In the autumn, yellow is the main environmental color, which is more suitable for the light-yellow background type. Green is the background color in summer, as dark colors are harder to find for natural enemies. The effect of the environmental temperature experienced during *H. axyridis* juvenile development on the adult phenotype has been fully demonstrated: a period of high temperature will affect the blackening degree of the structure [45]

Research has already found that these stains are the result of the comprehensive expression of a series of alleles [13]. It shows a mosaic dominant character, which is not stable inheritance [46]. Li Jiahui et al. reported that genetic diversity exists in almost every population, and a high rate of gene flow is also found in the population [47]. Some scholars also believe that the genetic inheritance of *H. axyridis* may be related to the genes controlling the color change of insects' body surface. However, so far, the internal causes of the color spot variation of the ladybird have not been fully understood.

*H. axyridis* adults aggregate in a fixed location throughout the winter in northeast China [48,49], and the pre-wintering or overwinter population can increase their long-term cold tolerance [43]. This potential to build up cold hardiness may have facilitated the ability of *H. axyridis* to invade other areas, and this property is also useful for long-term storage to apply this insect in pest control when needed. Indeed, our results showed that the survival ability of the overwinter *H. axyridis* population was greater than that of the experimental population (Figure 7), which had not previously been exposed to cold temperatures, demonstrating that the overwinter adults can be stored in a cold condition over the longer term. However, there was no difference in the survival rate between yellow and black adults of the same population. This may be due to the fact that the experimental adults were collected at the end of September. With the increase of low temperature treatment time, it is still very possible that there will be obvious differences in the survival rate of adult *H. axyridis* with different color spots, as reported previously [50]. Previous studies indicated a decline in survival of overwinter *H. axyridis* when stored at −5 °C and 10 °C [51], and the survival rate of overwinter *H. axyridis* adults collected from Jilin Agricultural University was more than 80% when stored at 3 °C and 6 °C for 150 days [50]. However, in our study, the survival rate of *H. axyridis* adults at day 60 of storage at 5 °C was less than 10% (Figure 7). This difference in the survival rate among studies suggests that the storage conditions, including feeding before storage, humidity and filler used, have an effect on survival beyond temperature.

Winter in temperate zones imposes significant environmental stress on arthropods [52,53]. Many studies have demonstrated that a cold acclimation process, especially at 0 °C and 5 °C, can significantly improve the cold tolerance of insects [54]. Consistent with these previous findings, we found that the contents of trehalose and glycogen in the overwinter *H. axyridis* population were higher than those of the experimental population, with a 10-fold increase in trehalose and an approximately 100-fold increase for glycogen (Figure 2). Similar results were reported for the insect *Ectomyelois ceratoniae*, with an accumulation of trehalose and total sugars from October to February of the next year [55], whereas the contents of trehalose and glycogen in November and March were higher than those of other months in the beetle *Pityogenes chalcographus* [56]. We found higher trehalose and glycogen contents in the overwinter population than in the experimental population for both yellow and black adults (Figure 2). These results indicated that most insects could accumulate trehalose and total sugar under exposure to a cold environment, although there appear to be exceptions. For example, the trehalose and total sugar content of overwinter *H. axyridis* population decreased along with the temperature increased. The total sugars and fat content decreased gradually with the prolongation of the storage time when the overwinter *H. axyridis* population was stored under cold conditions [57]. This may be because insects will accumulate a lot of substances in their bodies before overwintering, such as fats, proteins, carbohydrates, etc. However, these substances are gradually consumed with the prolongation of overwintering time.

Trehalase participates in and regulates both homeostasis and development, and is involved in blood sugar or energy metabolism in insects, including growth, stress recovery, flight metabolism and chitin synthesis during molting [20]. The activities of TRE1 and TRE2 jointly regulate the trehalose content balance and provision of energy to sustain insect physiological activities. The trehalose content may increase when the activity of two trehalose activities is decreased by way of knockdown of the *TRE* gene or injected trehalase inhibitor [19,58]. In our study, the activities of TRE1 and TRE2 commonly regulated the changes in trehalose content in the *H. axyridis* experimental population during cold storage, with some differences found in the activity of the two trehalases between yellow and black adults. The trehalose content along with the activities of both TRE1 and TRE2 decreased over the cold storage period in the overwinter population (Figure 3B,D). By contrast, a previous study showed that the content of trehalose and *TPS* expression in-creased significantly, whereas the activity of TRE1 decreased when *H. axyridis* experimental adults were exposed to a cold-stress condition with cooling from 25 °C to −5 °C [21]. Our study

indicated that the TRE1 activity played the key role in trehalose changes in the overwinter *H. axyridis* population during cold storage rather than the TRE2 activity, and that TRE1 can directly control the content changes of trehalose during the different stages of cold storage (Figure 3A,C). Disruption of the trehalose balance, either by knocking down the expression of *TRE* or *TPS* genes or suppressing the two types of trehalase activities, results in increased mortality of insects by blocking an important energy supply [59].

Insects of the order Coleoptera, including *H. axyridis* and *Tribolium castaneum*, have numerous trehalase genes [60], whereas most other insect species only have one soluble trehalase and one membrane-bound trehalase, such as *Spodoptera exigua* [57], *Apis mel-lifera* [61,62], *Bombyx mori* [63], *Laodelphax striatellus* [64], *Omphisa fuscidentalis* [35,65] and others [66], with the exception of *Nilaparvata lugens*, which has more than three trehalase genes [26,67]. *H. axyridis* has seven trehalase genes, which together determine and regulate the trehalase activity required for developmental and physiological processes [21]. TRE1 and TRE2 also appear to have different functions, with other *TREs* exhibiting complementary functions to compensate for them if another trehalase gene is inhibited [26]. We found different patterns of change in trehalase mRNA expression when *H. axyridis* adults were exposed to cold storage. *TRE1-1*, *TRE1-2*, *TRE1-5* and *TRE2-1* played key roles in regulating trehalase activity in the overwinter population (Figure 4), whereas nearly all seven trehalase genes (except for *TRE1-2*) contributed to TRE activity in the experimental population (Figure 5). The expression levels of the *GS* and *TPS* genes decreased during cold storage in the experimental population (Figures 5 and 6B), which was consistent with the significant decrease in the trehalose and glycogen contents. This demonstrated that trehalose and glycogen could not accumulate or that other sugars could not be transformed into trehalose during cold stress. However, we found that the content of trehalose increased while that of glycogen decreased during cold storage for the experimental *H. axyridis* adults. *TPS* gene expression was also found to increase under cold stress in a previous study [21], suggesting that glycogen can be transformed into trehalose during the cooling process. Our results also suggested that other sugars may first be used as energy sources in the first 10 days of cooling, followed by a rapid degradation of glycogen at 20 or 40 days, as reflected by the significant decrease in *GP* expression followed by its transformation to trehalose at 15 and 20 days as the trehalose was synthesized via *TPS* (with higher expression) in overwintering *H. axyridis* adults (Figure 6A). These results demonstrated the importance of trehalose for an anti-cold stress response, which needs to be maintained at a certain level even if other sugars can be transformed to trehalose.

Many studies have shown that the distribution of insect species is strongly affected by temperature, and overwintering insects need to adopt complex strategies to overcome the stressful environmental conditions [68]. In many cold tolerance studies, the super-cooling point (SCP) has been used as an important indicator to measure the strength of cold tolerance [69]. In Zhao's research, the SCP of the adults with a yellow bottom was lower than that of the adults with a black bottom during the whole wintering period, so the damage caused by freezing could be avoided by super-cooling the body fluid [40]. Moreover, rapid cold hardening and rapid cold exposure of a super-cooled state can increase the trehalose, carbohydrate energy and total energy content, as well as increase the survival of cells and the entire organism in *Belgica antarctica* [70,71]. As the temperature decreases gradually before winter, this provides a signal to insects, such as *H. axyridis*, to start accumulating sugars, including trehalose and glycogen. Lipids also accumulate under cold stress and acclimation, especially in female insects [51], because there are no food resources during the long winter. Indeed, we found a trend of a gradual increase in trehalose and glycogen content at the beginning of cold storage, followed by a decrease, suggesting that some other sugar or carbohydrate could transform into trehalose or glycogen in the early cold storage period of the overwinter *H. axyridis* population. This trend was also found for the yellow adults of the experimental population, which indicates that trehalose metabolism has an important regulating function in cold stress or cold storage. Moreover,

the higher survival rate for the overwinter *H. axyridis* population may have been related to the higher content of trehalose, as a previous study showed that a trehalose injection could increase the survival rate of *B. antarctica* larvae exposed at −15 °C and 30 °C for 3 h [72]. Further studies are needed to uncover the relationship of the increase in the contents of trehalose and other sugars via rapid cold hardening and rapid cold exposure with the cold resistance mechanism and the underlying genetic changes. Although the cold acclimation mechanism of insects is complex and multi-faceted, uncovering these details for *H. axyridis* can help to optimize its use as a natural insect predator [9], so that it could be stored over the long term to be released for pest control in the field at the time of need, with benefits to agriculture and forestry.

**Author Contributions:** Conceptualization, S.W. (Shigui Wang), Z.Z. and S.W. (Su Wang); methodology, Z.Z., S.W. (Shigui Wang) and S.W. (Su Wang); software, L.C., S.W. (Shasha Wang), W.Y. and Z.H.; validation, S.W. (Sijing Wan), J.H. and Z.S.; formal analysis, S.W. (Sijing Wan), J.H. and L.C.; investigation, Z.Z.; resources, S.W. (Su Wang); data curation, S.W. (Sijing Wan), J.H. and Z.S.; writing—original draft preparation, Z.Z., S.W. (Shigui Wang) and S.W. (Sijing Wan); writing—review and editing, S.W. (Sijing Wan); visualization, Z.S., W.Y. and Z.H.; supervision, S.W. (Shasha Wang) and W.Y.; project administration, S.W. (Shigui Wang); funding acquisition, Z.Z. and S.W. (Shigui Wang). All authors have read and agreed to the published version of the manuscript.

**Funding:** This work was supported by the National Key Research and Development Program of China (Grant no. 2017YFD0201000, 2022YFD1401204, 2022YFC2601405), the Hangzhou Science and Technology Development Program of China (Grant No. 20190101A01), the Technical Innovation Program of the Beijing Academy of Agriculture and Forestry Sciences (KJCX20200110), the Key Research and Development Program of Zhejiang Province, China (2021C02003) and the Demonstration and Extension Program of Science and Technology Achievements of Zhejiang Academy of Agricultural Sciences (tg2022008).

**Data Availability Statement:** Not applicable.

**Conflicts of Interest:** The authors declare no conflict of interest.

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
