# Peer review of "Regulatory Role of Trehalose Metabolism in Cold Stress of Harmonia axyridis Laboratory and Overwinter Populations"

_agronomy, doi:10.3390/agronomy13010148_

Round 1

Reviewer 1 Report

Dear Authors

The manuscript reports useful information.  However, the writing still needs work.  What I suggest to the authors is to read through every sentence and look for unnecessary words to delete. Kindly see my comments below.

Throughout the MS I found that scientific names of insects are written without authority. Kindly add authority along with insect names. For instance, Harmonia axyridis should be written as Harmonia axyridis Pallas

Line 18: Change “to compare”

Line 26: Change as “compared to black adults”

Line 32: Authority missing

Lines 57- 58: Add reference

Lines 71 – 73: Sentence difficult to follow try making it shorter

Lines 75 – 77: There is something not right about the sentence. Please consider rephrasing.

Lines 81 – 86: Very long sentence. Try breaking it down.

Line 93: Add “was used”

Line 96: Insect scientific name should be in Italics with authority

Lines 105: Insect scientific name should be in Italics with authority

Line 110: Change to “maintained”

Line 111: Insect scientific name should be in Italics with authority

Line 131: Did you mean to add this as a subheading?

Line 158: Change to “described”

Line 184: I would suggest changing animals to insects

Line 193: Change “to compared”

Line 194: Specify the years

Line 395: Change to “accumulation”

Line 428: Change to “numerous”

Lines 412- 413: Kindly consider rephrasing the sentence

Line 428: Add order next to coleoptera

Author Response

Dear reviewer:

Thank you for your hard work in reviewing our manuscript and your suggestions. All your suggestions are very important, and they are of great guiding significance for my paper writing and scientific research! Now we have carefully revised the manuscript according to your suggestions, and all revised chapters are marked in red. I will upload the modified specific information in the form of attachment.

Best regards!

Reviewer 2 Report

 Regulatory Role of Trehalose Metabolism in Cold Stress of Harmonia axyridis Laboratory and Overwinter Populations

Abstract

Line 18: Change  com-pared....to compared

Line 19-20: incomplete sentence....Yellow adults were dominant in the overwinter population. Why this mentioned here what about lab population

Line 26: cold storage compare black adults....it should be compared to

Abstract needs refinement

Introduction

Line 36: it has now spread into all environments???? It should be rewritten not all environment ..instead it has spread widely in ...(mention few prominent hosts)..

Materials and Methods

Line 96: italics Aphis medicaginis

Line 104: The mating pair expt shows the authors kept 5-8 boxes what I understood is these are replications, if so, then it should very clear about no. of boxes so don’t give range here. Besides, in bracket mention replications. Mention age of the adults

Line 105: itlics A. medicaginis follow this throughout your MS

Line 107: replace lavae with grubs

Line 110: remove hyphen in the word maintained

Line 114-119: Survey part must be clear ..how many plants observed, host, sample size, type of survey and area etc

Line 121-129: sexes of the beetles not mentioned. What stage they overwinter?

Line 158: remove hyphen de-scribed again in line 168, 174, 193, 228 as well.

I don’t understand the why mating experiment was carried out as no results and discussion on this.

Colour vs gene expression vs season may be highlighted in results and discussion

Author Response

(The authors gave the same response as above.)

Reviewer 3 Report

Interesting research, especially because it contributes to additional knowledge of this predator species.

Please check the reference list.

Author Response

(The authors gave the same response as above.)

Round 2

Reviewer 2 Report

Line 118-124: Survey part must be clear ..how many plants observed, host, sample size, type of survey and area etc. Still it is not clear Line 129-130: sexes of the beetles not mentioned. What stage they overwinter? I don’t understand the why mating experiment was carried out as no results and discussion on this. Colour vs gene expression vs season may be highlighted in results and discussion Most of the comments are not attempted mainly the above need to addressed otherwise appropriate rejoinder may be given

Author Response

Dear reviewer:

Thank you again for your hard work in reviewing our manuscript and your suggestions. All your suggestions are very important, and they are of great guiding significance for my paper writing and scientific research! Now we have carefully revised the manuscript according to your suggestions, and all revised chapters are marked in red. I will upload the modified specific information in the form of attachment.

Best regards!
